# Microstructure, Properties, and Titanium Cutting Performance of AlTiN–Cu and AlTiN–Ni Coatings

**Jiyong Yi [1],\*, Kanghua Chen [2],\* and Yinchao Xu [2]**

[1]  School of Physical and Mechanical Engineering, Jishou University, Jishou 416000, China
[2]  Science and Technology on High Strength Structural Materials Laboratory, Central South University, Changsha 410083, China; xuchao-100@163.com
\*  Correspondence: yijiyong@jsu.edu.cn (J.Y.); khchen@csu.edu.cn (K.C.)

**Abstract:** In this study, three kinds of coatings, AlTiN, AlTiN–Ni, and AlTiN–Cu were deposited via the cathodic arc evaporation method. The microstructure, mechanical properties, oxidation resistance, and cutting behavior of these coatings were then investigated. The incorporation of Cu(Ni) into AlTiN eliminated its columnar structure and led to an increase in the growth defects of its macroparticles. The addition of Cu and Ni decreased the hardness of the coatings, their elastic moduli, and their friction coefficients. All of the AlTiN, AlTiN–Ni, and AlTiN–Cu coatings presented sufficient adhesion strength values. The oxidation resistance of these three coatings was determined to be in the following order: AlTiN > AlTiN–Ni > AlTiN–Cu. Titanium turning experiments indicated that the cutting force was reduced and the tool life was improved through doping with Cu(Ni) elements, dependent on cutting speed. The AlTiN–Ni coating showed the best performance at a high cutting speed, whereas the AlTiN–Cu coating was more successful at a lower cutting speed.

**Keywords:** AlTiN–Ni coating; AlTiN–Cu coating; oxidation resistance; titanium cutting

## 1. Introduction

Titanium is difficult to machine for many reasons. Its low thermal conductivity creates extremely high temperatures at the tool-chip interface, its high chemical affinity results in severe adhesion to cutting tools, and its high plasticity and toughness give rise to poor cutting stability [1–5]. Tools can be protected by a coating to block the chemical reaction between the tool and workpiece, which can inhibit the generation of cracks during the cutting process and can reduce the occurrence of fractures [6]. Recently, TiAlN coatings have been widely used in cutting tools as wear-resistant protection due to their high chemical stability, excellent oxidation resistance, and high wear resistance [7,8]. However, TiAlN coatings prepared by conventional methods often exhibit a columnar structure and have a high friction coefficient. These properties make them susceptible to cracks and severe adhesion wear during the machining of Ti [9,10]. Therefore, it is important to improve the structure and properties of TiAlN coatings so as to accommodate the machining of Ti alloys.

One promising avenue of research is adaptive lubricious coatings. These adaptive coatings have been characterized as having a low friction coefficient during tribo-tests, which leads to a low wear rate [11–14]. The incorporation of Cu(Ni) into TiAlN coatings has also been studied. Cu(Ni) forms as crystals, rather than dissolving into the TiAlN matrix because Cu(Ni) is mutually immiscible with TiAlN and because its nitride is thermally unstable [15]. TiAlN coatings doped with Cu (Ni) have been shown in previous studies to eliminate the columnar crystal structure, and to both decrease a coating's friction coefficient and increase its toughness [15–21]. Moreover, Blov et al. found that TiAlN coatings doped with Cu and Ni exhibit excellent cutting performance during both the continuous and intermittent turning of steel [15]. Another effective method involves improving the oxidation

resistance of coatings [9]. A single-phase cubic TiAlN coating with a high Al content (referred to as AlTiN) has been shown to exhibit excellent mechanical properties and oxidation resistance. It is likely possible to enhance cutting performance for hard-to-machine materials by improving coatings' oxidation resistance and reducing their friction coefficient simultaneously. According to previous studies, an AlTiN–Cu(Ni) coating with ~1.5% Cu(Ni) can exhibit excellent mechanical properties and can demonstrate an excellent cutting performance [22,23]. However, little research has been directed towards the contrasts between AlTiN, AlTiN–Ni, and AlTiN–Cu coatings. In particular, the oxidation resistance and titanium turning behavior of these coatings have not yet been explored. In this study, therefore, AlTiN and AlTiN–Cu(Ni) coatings were fabricated by the cathodic arc evaporation. The microstructures, properties, and cutting performances of AlTiN, AlTiN–Ni, and AlTiN–Cu coatings were then investigated. In particular, the mechanism of Cu(Ni)-action on the oxidation resistance and titanium cutting behavior of AlTiN coatings was explored.

## 2. Experimental Details

### 2.1. Coating Deposition

The AlTiN, AlTiN–Ni, and AlTiN–Cu coatings were deposited on a cemented carbide substrate (WC-6 wt.% Co) and polycrystalline $Al_2O_3$ using a cathodic arc evaporation system with $Ti_{0.33}Al_{0.67}$, $(Ti_{0.33}Al_{0.67})_{0.985}Ni_{0.015}$, and $(Ti_{0.33}Al_{0.67})_{0.97}Cu_{0.03}$ powder metallurgy targets, respectively. Prior to the deposition, the substrates were cleaned via Ti-ion-etching at a bias voltage of −900 V for 20 min. During the deposition, the substrate bias voltage was −80 V, the cathode current was 100 A, the deposition temperature was 500 °C, the working gas pressure in the $N_2$ atmosphere was ~2.2 Pa, and the deposition time was 3 h.

### 2.2. Coating Characterization

Transmission electron microscopy (TEM, JEM-2100F, Tokyo, Japan) was employed in combination with focused ion beam microscopy (FIB, Helios Nanolab 600i, Thermo Fisher Scientific, Waltham, MA, USA) to characterize the microstructure of the coating. The chemical composition of the coatings was analyzed using electron probe microanalysis (EPMA, JXA-8530F, Tokyo, Japan). The phase and crystal structure of the as-deposited coatings on the cemented carbide were evaluated using X-ray diffraction (XRD, D/max2550pc, Rigaku Co., Ltd., Tokyo, Japan) with Cu Kα radiation. The nanohardness H of the coatings was determined using a nanoindentation test (OPX, CSM, Geneva, Switzerland) with a Berkovich diamond tip. During the nanoindentation test, the loading rate was 20 mN/min, the maximum load was 10 mN, and the loading time was 15 s. The adhesion of the coating was assessed using a scratch tester (MST, CSM, Geneva, Switzerland) with an indenter tip of radius 50 μm, maximum load of 30 N at a loading rate of 30 N/min, and scratch length of 2 mm. To test the oxidation resistance behavior of the coating, the coated polycrystalline $Al_2O_3$ substrates were treated using a tube furnace in the ambient atmosphere.

### 2.3. Cutting Experiments

The cutting forces were measured using a three-component piezo-electric dynamometer (KISTLER 9257B, Winterthur, Switzerland). A charge amplifier (KISTLER 5080, Winterthur, Switzerland), a graphical programming environment (Dyno Ware, 2825A) and data acquisition hardware (PCIM-DAS1602/16) were used for data acquisition and data analysis. The cutting data for the experiments were used to continuously Titanium (TC4) with a cutting speed ($v_c$) of 40, 60, and 80 m/min, a depth of cut ($a_p$) of 0.5 mm, and feed rate (f) of 0.2 mm prerevolution (mm/r).

Tool wear tests were carried out on a CNC lathe CK7525 machine under turning conditions during the wet machining of titanium (TC4). The continuous turning of titanium with VNEG120408-NF type carbide was conducted at cutting speed ($v_c$) 60 and 80 m/min, feed rate (f) of 0.2 mm per revolution (mm/r), and 1.0 mm depth of cut ($a_p$). Optical microscopy (EV3020, Eassaon, Ningbo, China) was used

to examine the wear and average flank wear land width (*Vb*) of the coated tools. The criterion for the tool life-time is when the flank wear lands exceed 0.3 mm.

## 3. Results and Discussion

### 3.1. Microstructure and Composition

The chemical compositions of the $Al_xTi_{1-x}N$, $Al_xTi_{1-x-y}N–Ni_y$, and $Al_xTi_{1-x-y}N–Cu_y$ coatings, as analyzed using EPMA, are $Al_{62.5}Ti_{37.5}N$ (named AlTiN), $Al_{62.8}Ti_{35.7}N–Ni_{1.5}$ (named AlTiN–Ni), and $Al_{63.1}Ti_{35.5}N–Cu_{1.3}$ (named AlTiN–Cu), respectively. The XRD patterns of the AlTiN, AlTiN–Ni, and AlTiN–Cu coatings, as deposited onto the cemented carbide substrate, are shown in Figure 1. Their single-phase centered cubic structures are shown. The AlTiN coating behaved between c-TiN (PDF#65-5759) and c-AlN (PDF#25-1495) due to the fact that the solid solution of Al had a smaller atomic radius. Following the addition of Ni and Cu, the (111) peak intensity sharply decreased and the (200) peak was broadened. These observations indicate a decrease in grain size and a weakening of the coating's texture [24]. WC peaks were also observed, as shown in Figure 1.

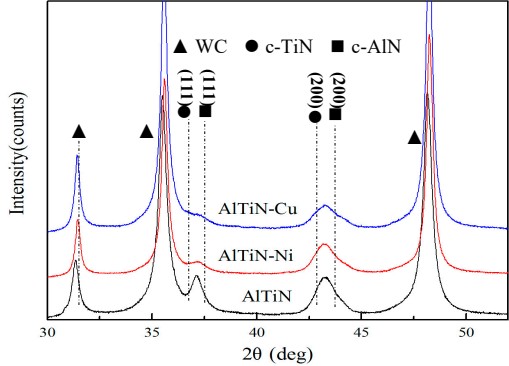

**Figure 1.** XRD patterns of AlTiN, AlTiN–Ni, and AlTiN–Cu coatings.

Figure 2 shows SEM fracture cross-sections for the AlTiN, AlTiN–Ni, and AlTiN–Cu coatings, deposited on the cemented carbide substrate. All of the coatings show dense structures, each with a coating thickness of ~3.0 μm. As indicated in Figure 2a, columnar grains are evident in the pure AlTiN coating, and these grains increase in size throughout the coating. The addition of Cu and Ni to the AlTiN coating (in Figure 2b,c) results in the transition from a continuous columnar grain structure into a comparatively smooth structure. To gain detailed information on the microstructure of the AlTiN–Ni and AlTiN–Cu coating, TEM fracture cross-sections and the selected area diffraction (SAED) results were examined, as shown in Figure 3. According to previous studies [9,24], the TEM and SAED results of the single-phase cubic AlTiN coating exhibit a columnar growth structure and spotty rings, respectively. As seen in Figure 3, the AlTiN–Ni and AlTiN–Cu coatings exhibit a discontinuous columnar growth structure, and the SAED results show that the AlTiN–Ni and AlTiN–Cu coatings have continuous rings. This indicates that grain size decreased with the addition of Ni and Cu. This can be attributed to the impurities in metal Cu(Ni) hindering grain growth. This stimulates the re-nucleation of grains during the coating deposition [22–24]. Myung et al. [20] found that ~1.5 at.% Cu is sufficient to form a dense nanocrystal TiN–Cu nanocomposite coating without a columnar structure. Chen et al. [24] have also reported similar results.

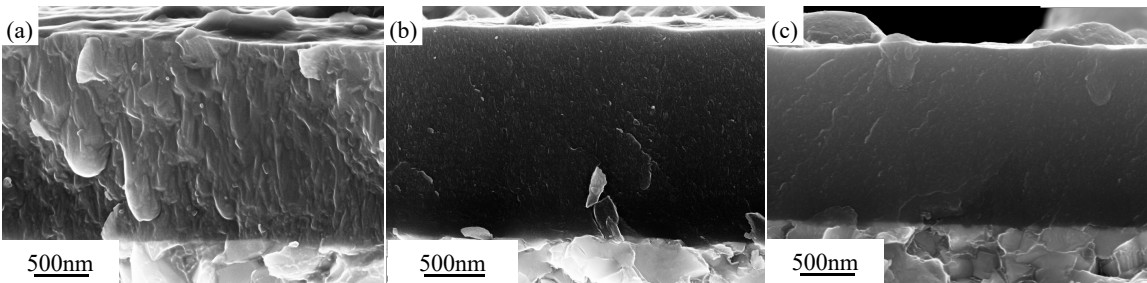

**Figure 2.** SEM morphologies of (**a**) AlTiN, (**b**) AlTiN–Ni, and (**c**) AlTiN–Cu coatings.

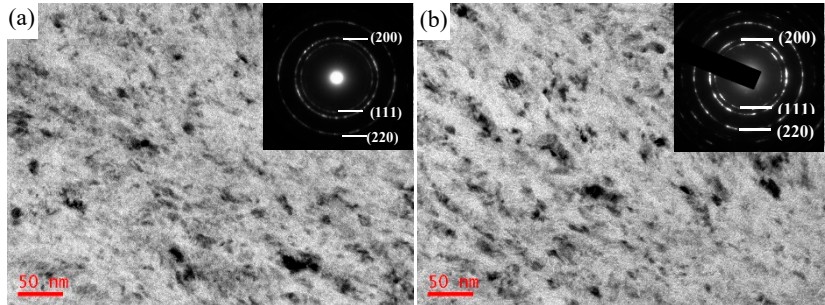

**Figure 3.** TEM morphologies of (**a**) AlTiN–Ni and (**b**) AlTiN–Cu coatings.

Surface defects in metallic macroparticles (MPs) are inevitable when cathodic arc evaporation deposition technology is used, and they often cause the properties of the coatings to deteriorate [25,26]. Figure 4 shows SEM images of the surface morphologies of the AlTiN, AlTiN–Ni, and AlTiN–Cu coatings. The MP density is defined as $f_{MPs} = A_{MPs}/A_{total}$ [26], where $A_{MPs}$ and $A_{total}$ are the area of all the metallic MPs in the micrograph and the total area, respectively. As seen in Figure 4, the number of MPs increased following the addition of Ni and Cu. Compared to the AlTiN–Ni coatings, more MPs appeared on the surface of AlTiN–Cu coatings. The MP densities of the AlTiN, AlTiN–Ni, and AlTiN–Cu coatings, deposited via cathodic arc evaporation, are 0.057, 0.105, and 0.188, respectively.

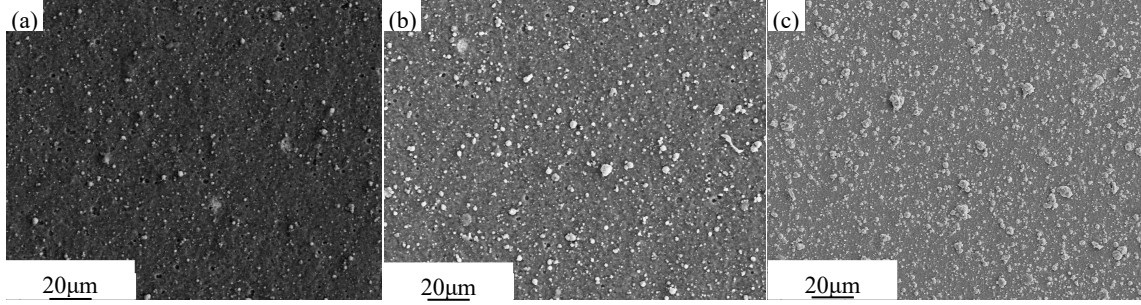

**Figure 4.** Surface morphologies of (**a**) AlTiN, (**b**) AlTiN–Ni, and (**c**) AlTiN–Cu coatings.

*3.2. Mechanical Properties*

Table 1 presents the mechanical properties of the AlTiN, AlTiN–Ni, and AlTiN–Cu coatings. The nanohardness and elastic modulus of the AlTiN–Ni (24.3 GPa and 315.8 GPa, respectively) and AlTiN–Cu coating (23.7 GPa and 300.2 GPa, respectively) are slightly lower than those of the AlTiN coating (26.1 GPa and 332.2 GPa, respectively). This is mainly because the former contains soft Cu and Ni. This result is in agreement with the findings of Chen et al. [24]. Table 1 also presents scratch test results for the AlTiN, AlTiN–Ni, and AlTiN–Cu coatings. The adhesion strength indicates the complete peeling of the coatings from the cemented carbide substrate. As shown in Table 1 and Figure 5, the adhesion strengths of the AlTiN, AlTiN–Ni, and AlTiN–Cu coatings are 18.9 N, 18.3 N, and 18.7 N, respectively. All of the AlTiN coatings exhibited sufficient adhesion strength. The slightly

lower adhesion strength of the AlTiN–Ni and AlTiN–Cu coatings can be attributed to their higher MP contents, which results in decreased surface roughness [27].

**Table 1.** Mechanical properties of AlTiN, AlTiN–Ni, and AlTiN–Cu coatings.

| Coatings | H (GPa) | E (GPa) | Adhesion Strength (N) |
|---|---|---|---|
| AlTiN | 26.1 ± 2.9 | 337.5 ± 14.3 | 18.9 ± 1.1 |
| AlTiN–Ni | 24.3 ± 1.8 | 315.8 ± 11.9 | 18.3 ± 1.0 |
| AlTiN–Cu | 23.7 ± 1.7 | 300.2 ± 11.2 | 18.7 ± 1.3 |

The friction coefficient was measured as the ratio of the tangential force to the normal force. The measured friction coefficients of the AlTiN, AlTiN–Ni, and AlTiN–Cu coatings obtained from the scratch experiments are shown in Figure 5. Experimentally, the curves of the friction coefficients are composed of three parts: the beginning part correlates with the loading stage, the second part correlates to steady-state scratching, and the third part of the coating correlates to the beginning of flanking and the complete peeling off from the substrate. The effective values were obtained by calculating the average values of the second parts. It is obvious that the measured friction coefficients of the AlTiN–Cu and AlTiN–Ni coatings are lower than that of the AlTiN coating. The metallic Ni and Cu phases in the AlTiN–Cu and AlTiN–Ni coatings may provide lubricant effects during the sliding process [15].

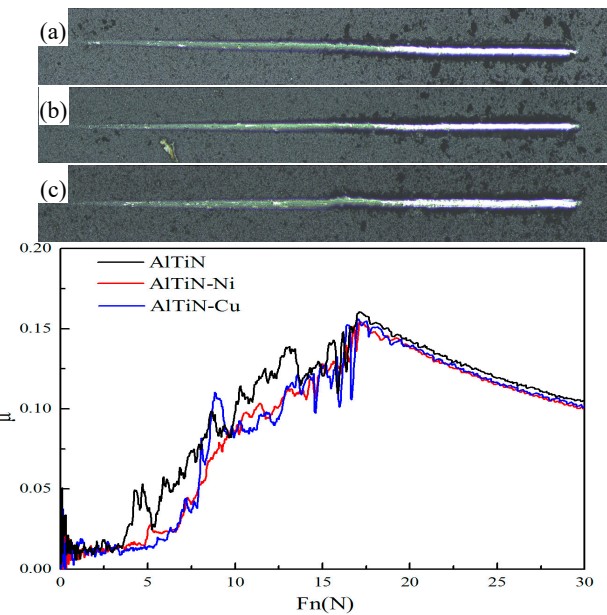

**Figure 5.** Images of a scratch on the surface and friction coefficient of (**a**) AlTiN, (**b**) AlTiN–Ni, and (**c**) AlTiN–Cu coatings.

*3.3. Oxidation Resistance*

The oxidation resistance values of the coatings are closely related to their service behavior regarding cutting applications. Figure 6 shows SEM fracture cross-sections of the AlTiN, AlTiN–Cu, and AlTiN–Ni coatings on polycrystalline $Al_2O_3$ substrates, after isothermal oxidation at 800 °C for 2 h in a tube furnace. The AlTiN coating exhibited the best oxidation resistance, with oxide scales of ~0.19 μm. The oxidation resistance of the AlTiN–Ni coating was close to the value of AlTiN coating, with oxide scales of ~0.20 μm. The AlTiN–Cu coating had the lowest oxidation resistance, with the oxide scales of ~0.27 μm. In general, grain refinement has been shown to be effective in enhancing the oxidation resistance of coatings [28]. However, here, alloying with Cu and Ni leads to a drop in oxidation resistance owing to the additive Ni(Cu) significantly increasing the growth defects of MPs on the surfaces of AlTiN coatings. The growth defects of MPs on the surfaces of the coatings have

an unfavorable effect on oxidation resistance, due to the resulting increase in the oxidation diffusion process [29]. When the temperature was increased to 900 °C (Figure 7), a similar case occurred. However, the oxide scales were increased, due to the higher oxidation temperature.

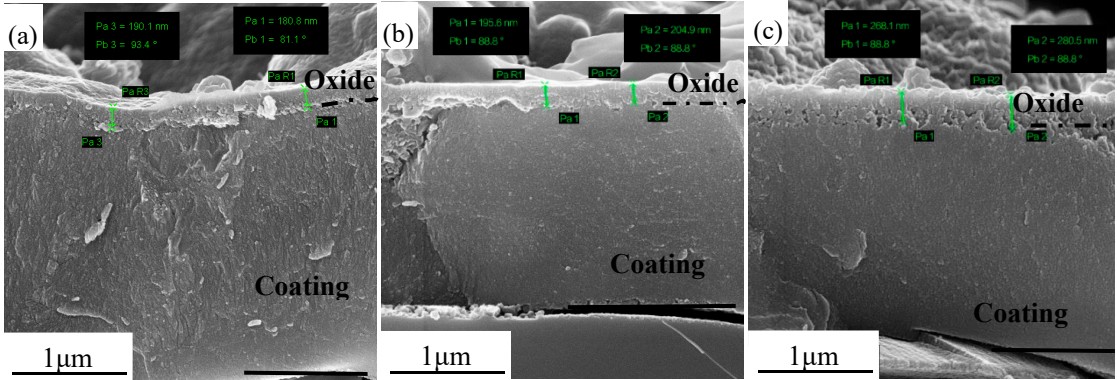

**Figure 6.** SEM fracture cross-section images of (**a**) AlTiN, (**b**) AlTiN–Ni, and (**c**) AlTiN–Cu coatings on polycrystalline $Al_2O_3$ after isothermal oxidation at 800 °C for 2 h.

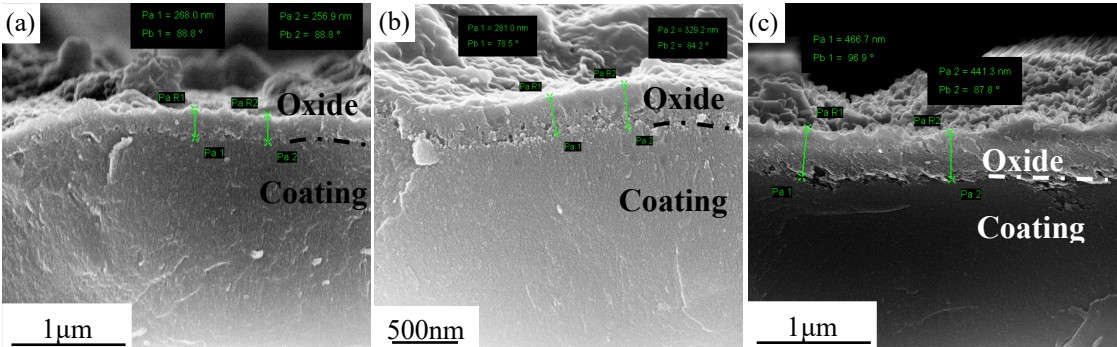

**Figure 7.** SEM fracture cross-section images of (**a**) AlTiN, (**b**) AlTiN–Ni, and (**c**) AlTiN–Cu coatings on polycrystalline $Al_2O_3$ after isothermal oxidation at 900 °C for 1 h.

*3.4. Cutting Experiments*

Figure 8 shows the cutting forces of the AlTiN, AlTiN–Ni, and AlTiN–Cu coated tools when turning TiC4 alloys, with cutting speeds of 40, 60, and 80 m/min. As seen in Figure 8, the cutting force (Fx, Fy, and Fz) decreased following the addition of Cu and Ni, suggesting that the soft metallic Cu and Ni could effectively reduce the cutting force. The AlTiN–Cu coated tools possessed the lowest cutting force when turning Ti at various cutting speeds, out of the three kinds of coated tools. Figure 9 shows the average flank wear in relation to the cutting time, during the turning of Ti at cutting speeds of 60 m/min and 80 m/min. The coated carbide tool reached its tool life after the average flank wear land width (Vb) reached 0.3 mm. As seen in Figure 9a, the cutting lifetime of the uncoated tool was the shortest, at ~20 min. It is obvious that all the coated tools possessed better wear resistance than the uncoated tools. This is because the coating reacts at high temperatures to form protective alumina tribo-films, which could effectively reduce the intensive interaction between the tool and the workpiece material [10]. The cutting lifetime of the AlTiN-coated tool was as high as 28 min at a cutting speed of 60 m/min. The AlTiN–Ni coatings retarded the tool wear, compared to the AlTiN-coated tools. The AlTiN–Cu-coated tool possessed the longest tool life (over 38 min). The superior cutting performance of the AlTiN–Cu-coated tool at the cutting speed of 60 m/min can be attributed to its lower cutting force [30]. When the cutting speed was increased to 80 m/min, the AlTiN–Ni-coated tool exhibited the longest cutting lifetime. The cutting lifetime decreased to 10 min due to the harsher working conditions caused by the high cutting speed. The good machining performance of AlTiN–Ni

coating at high cutting speeds can be attributed to its balance of a low cutting force and a high oxidation resistance.

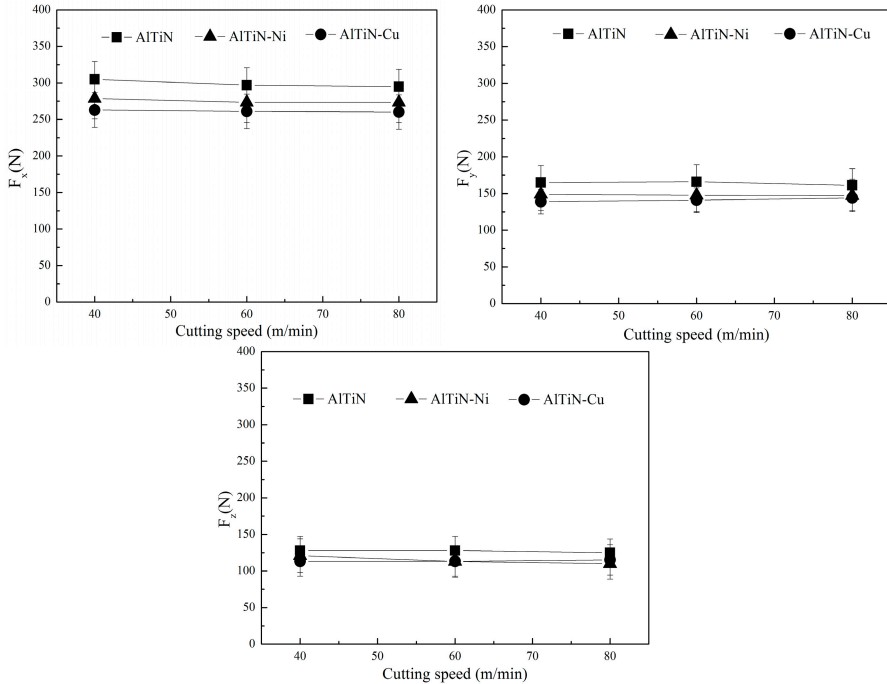

**Figure 8.** Cutting force of AlTiN, AlTiN–Ni, and AlTiN–Cu coated tools turning TiC4 alloys at various cutting speeds.

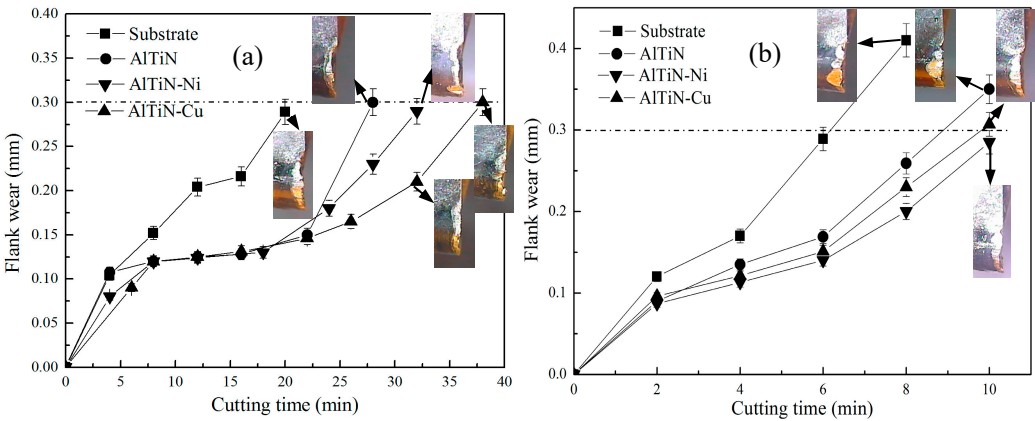

**Figure 9.** The flank wear versus cutting time at different cutting speeds: (**a**) 60 m/min and (**b**) 80 m/min.

SEM images and the energy-dispersive X-ray spectroscopy (EDS) results of the uncoated, AlTiN, AlTiN–Ni, and AlTiN–Cu-coated tools were obtained to further investigate the wear behavior of the tools during turning of the TC4 alloys; the results are shown in Figures 10 and 11, respectively. As shown in Figures 10b and 11a, the white areas mainly consist of tungsten (W). This indicates that the AlTiN coating in this area has been removed. Points C, D, and E (in Figure 11c–e) mainly contain Al, Ti, and N, which means that the coating in this area is still present. The EDS results of the adhesive bonds (Point B) mainly indicate the presence of Al, Ti, O, and V. This indicates that these adhesive bonds only come from the TC4 workpiece [9]. The main wear modes of the uncoated, AlTiN, AlTiN–Ni, and AlTiN–Cu-coated carbide tools are adhesion and chipping.

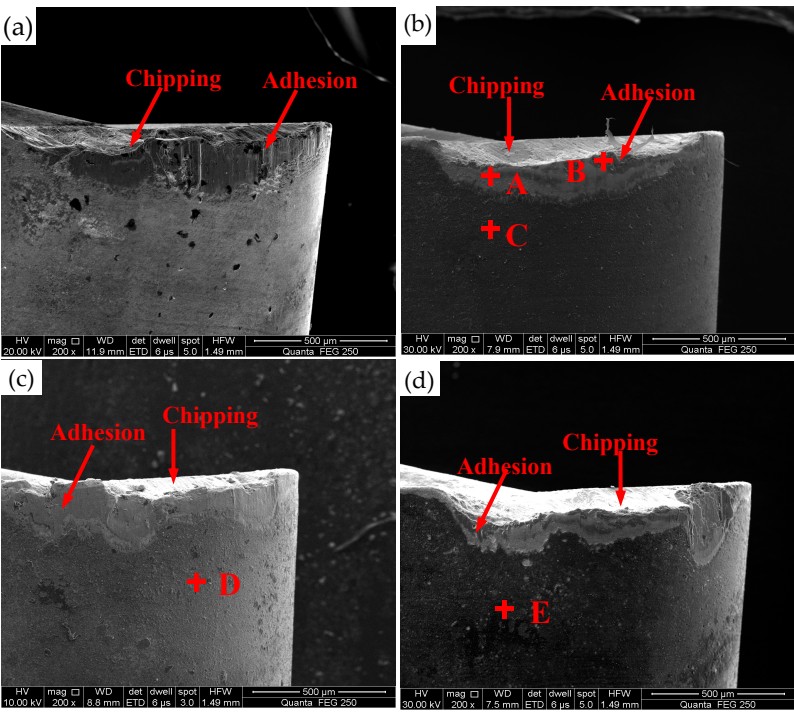

**Figure 10.** SEM images of the coated tools after TC4 turned at 60 m/min: (**a**) uncoated, (**b**) AlTiN, (**c**) AlTiN–Cu, and (**d**) AlTiN–Ni.

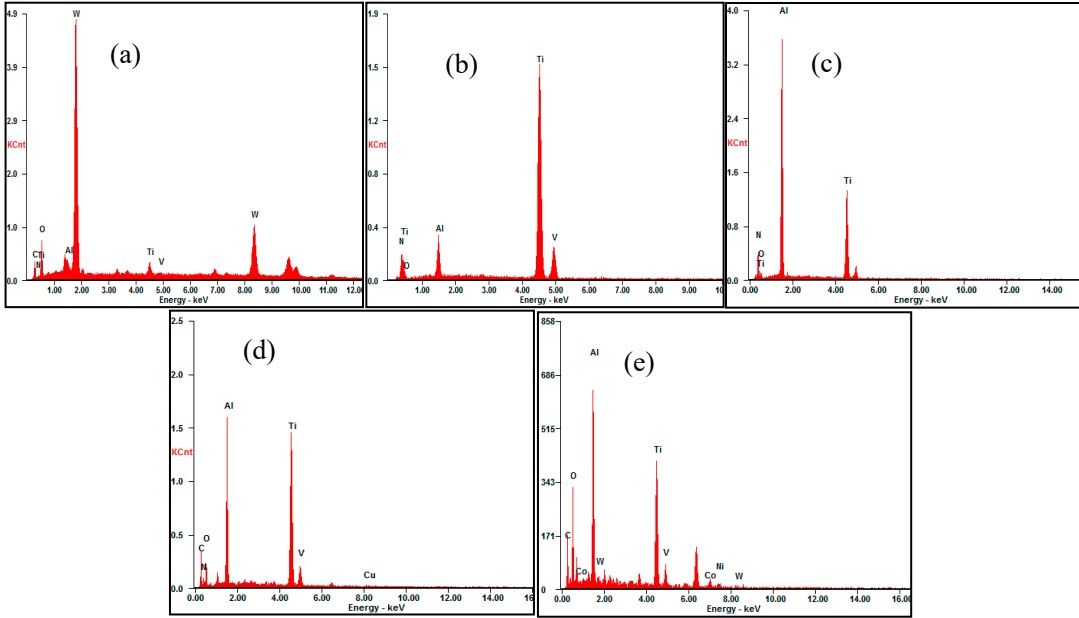

**Figure 11.** Energy-dispersive X-ray spectroscopy (EDS) results of the coated tools after TC4 turned at 60 m/min: (**a**) Point A, (**b**) Point B, (**c**) Point C, (**d**) Point D and (**e**) Point E.

## 4. Conclusions

Here, AlTiN, AlTiN–Ni, and AlTiN–Cu coatings were synthesized via the cathode arc evaporation method. The addition of Cu(Ni) to the AlTiN coating resulted in a decrease in its grain size and hardness. Adding Cu(Ni) into the AlTiN coating led to an obvious increase in the growth defects of MPs in the coating, and thereby reduced oxidation resistance. The following ordering of the three coatings regarding oxidation resistance was established: AlTiN > AlTiN–Ni > AlTiN–Cu. Titanium turning experiments showed that the addition of Cu(Ni) effectively decreased the cutting force, thereby

extended tool lifetimes at various cutting speeds. The longest tool lifetime was exhibited by the AlTiN–Cu-coated tool due to it exhibiting the lowest cutting force, at a cutting speed of 60 m/min. However, the AlTiN–Ni-coated tools exhibited a higher tool lifetime than AlTiN–Cu-coated tools at cutting speeds of 80 m/min. This is related to the balance of cutting force and oxidation resistance.

**Author Contributions:** Conceptualization, J.Y. and K.C.; methodology, J.Y. and Y.X.; investigation, J.Y., Y.X. and K.C.; resources, J.Y., Y.X. and K.C.; data curation, J.Y. and Y.X.; writing—original draft preparation, J.Y. and K.C.; writing—review and editing, J.Y.; visualization, J.Y. and Y.X.; project administration, K.C.; funding acquisition, J.Y. and K.C.

**Funding:** This research was funded by the National Natural Science Foundation of China, grant number 51327902" and "The Fundamental Research Funds for the Central Universities of Central South University, grant number 2017zzts106.

**Acknowledgments:** The authors gratefully acknowledge the financial support of the major research equipment development projects of National Natural Science Foundation of China (Grant No. 51327902), and the Fundamental Research Funds for the Central Universities of Central South University (Grant No. 2017zzts106).

**Conflicts of Interest:** This manuscript has not been published or presented elsewhere in part or in entirety and is not under consideration by another journal. We have read and understood your journal's policies, and we believe that neither the manuscript nor the study violates any of these. There are no conflicts of interest to declare.

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
