# Peer review of "Microstructure, Properties, and Titanium Cutting Performance of AlTiN–Cu and AlTiN–Ni Coatings"

_coatings, doi:10.3390/coatings9120818_

Round 1

Reviewer 1 Report

After reading the manuscript, I have the following comments:

1. Extend Chapter 1 with an analysis of the results of other machining tests with the AlTiN-Ni and AlTiN-Cu coated tools.
2. References in the text are wrong formatted or missing.
3. Lines 81-82 are analogous to table 1, you specifies ap = 1 and 0.5 mm.
4. Figure 2, 5, 6 - invisible or hidden descriptions.
5. Figure 3, 5, 6, 10 - the descriptions fonts size is too small.
6. Line 159 and others - you analyze the cutting force components, not the cutting force. Instead of Fx, Fy, Fz cutting force components use feed, radial and tangential cutting force components respectively. Do you analyze the average values of cutting force components? Explain this and add error bars in Figure 7.

Author Response

Extend Chapter 1 with an analysis of the results of other machining tests with the AlTiN-Ni and AlTiN-Cu coated tools.

Response: We added the machining tests with the AlTiN-Ni and AlTiN-Cu coated tools in chapter1

References in the text are wrong formatted or missing.

Response: We have modified the wrong formatted or missing references in the text

Lines 81-82 are analogous to table 1, you specifies ap = 1 and 0.5 mm.

Response: We have removed Table 1. The specifies ap = 1 and 0.5 mm are investigated cutting force and cutting lifetime respectively.

Figure 2, 5, 6 - invisible or hidden descriptions.

Response: We have modified the Figure 2, 5, 6 - invisible or hidden descriptions

Figure 3, 5, 6, 10 - the descriptions fonts size is too small.

Response: We have modified the descriptions of Figure 3, 5, 6, 10

Line 159 and others - you analyze the cutting force components, not the cutting force. Instead of Fx, Fy, Fz cutting force components use feed, radial and tangential cutting force components respectively. Do you analyze the average values of cutting force components? Explain this and add error bars in Figure 7.

Response: The Fx, Fy, Fz cutting force is the average values. We added error bars in Figure 7.

Reviewer 2 Report

This paper compares various mechanical properties such as nano hardness, elastic modulus and adhesion strength of AlTiN, AlTiN-Cu and AlTiN-Ni coatings

Also, author performed many experiments such as XRD, SEM, TEM, cutting test.

I have some comments for publication to this journal as written below.

Major

1) In this paper, author put the word oxidation resistance in the title and key words. However, there is not much mention of oxidation resistance in the paper, and oxidation resistance did not increase with the addition of Cu and Ni. So, the word ‘oxidation resistance’ in the title and key words seems to be confused with the data in the manuscript.

2) In Figure 2

2-1. author mentioned and marked the columnar structure, but the crystal plane is not visible. The crystal plane is required to explain the author’s all opinion. So author should add it or re-analyze.

2-2. author expressed ‘indicating an obviously decreasing grain size ~’. However, The TEM images doesn’t look like to satisfy author’s mentions. Therefore, the grain should be marked in the figure like Figure 2(a) or other data such as SEAD pattern should be required.

2-3. It is not clear that decrease in grain size is resulted from the dopants, even if authors show the other TEM images. So I would like to recommend TEM-EDS. It would be the clear proof.

3) In Figure 3.

3-1. Author used the expression ‘the number of macroparticle increased with the addition of Ni and Cu’, but the difference does not seem obviously. So, it would be better to plot the numbers through a histogram. And also, it is helpful to compare the macroparticle size of AlTiN, AlTiN-Cu, AlTiN-Ni coatings, if the particle size is displayed in the figure.

4) In Figure 4.

4-1. As my knowledge, the friction coefficient is deeply related to surface roughness. When Cu and Ni are doped, the surface roughness should be increased due to increase in macro particles. However, the friction coefficient is decreased after adding the Cu and Ni. I think that it is ironical results, so it is necessary to explain why the lubricant effect occurs.

4-2. The author mentioned that the friction coefficient graph divided into three parts. However, it is not clear where the start point of the parts.

4-3. Authors mentioned that ‘The effective values are obtained by calculating the average values of the second parts’. If the second part is from 5 N to 18N, it is wrong. This part is still in error bar and they are not stable. Or if the second part is from 18N, authors need to show zoom-in image, because it is the proof of the author’s mention (line 137, The metallic~). And the difference between red and blue lines is not visible.

5) In figure 8, Authors mentioned that ‘The good machining performance of AlTiN-Ni coating at high cutting speed is attributed to its relative low cutting force and high oxidation resistance’. However, in Figure 7. (a) and (b), AlTiN-Cu cutting force is lower than AlTiN-Ni and oxidation resistance of AlTiN is better than AlTiN-Ni. I think it is ironical results, so authors need explanation of relation between machining performance, cutting force and oxidation resistance.

6) In figure 9 and 10, SEM images and EDS results of uncoated specimens and AlTiN coated specimens for investigating the wear behavior. However, the author’s paper is about AlTiN-Cu and AlTiN-Ni coatings, so I think that these results are irrelevant. And also if authors want to study about it, it is appropriate to add SEM images or EDS results of AlTiN-Cu and AlTiN-Ni coated specimens, as well.

7) In conclusion, Author mentioned the word ‘nanocomposite’. However, we cannot find any relevant data to explain the nano-scale. TEM images as my comment No.2 are still unclear for the word ‘nano’. And I don’t think that it is nanocomposite, because of XRD pattens and macroparticles in SEM images. If authors want to insist that it is nanocomposite, nano-scale analysis is obviously required, such as clear TEM images.

Minor

1) The 144,145th sentences are considered to be the contents of the experiment, not the result and discussion.

2) In figure 5 and 6, Letters in the black box are not visible, and the word Oxide and Coating are cut off or do not match the color.

3) In figure 7, (a), (b) and (c) are cut off in the figure, and it would better to display and error bar in each figure.

4) Author needs a full inspection of the figure once again.

Author Response

1) In this paper, author put the word oxidation resistance in the title and key words. However, there is not much mention of oxidation resistance in the paper, and oxidation resistance did not increase with the addition of Cu and Ni. So, the word ‘oxidation resistance’ in the title and key words seems to be confused with the data in the manuscript.

Response: We have changed the name of the article to “microstructure, properties and titanium cutting performance of AlTiN-Cu and AlTiN-Ni coating.

2) In Figure 2

2-1. author mentioned and marked the columnar structure, but the crystal plane is not visible. The crystal plane is required to explain the author’s all opinion. So author should add it or re-analyze.

Response: We added the Fig2 and re-analyzed

2-2. author expressed ‘indicating an obviously decreasing grain size ~’. However, The TEM images doesn’t look like to satisfy author’s mentions. Therefore, the grain should be marked in the figure like Figure 2(a) or other data such as SEAD pattern should be required.

Response:We add the SEAD pattern in Fig 3

2-3. It is not clear that decrease in grain size is resulted from the dopants, even if authors show the other TEM images. So I would like to recommend TEM-EDS. It would be the clear proof.

Response:combined the XRD, SEM cross-section, TEM, SEAD and previous results, It would be the clear proof that the grain size is decrease with the addition of Cu(Ni).  

3) In Figure 3.

3-1. Author used the expression ‘the number of macroparticle increased with the addition of Ni and Cu’, but the difference does not seem obviously. So, it would be better to plot the numbers through a histogram. And also, it is helpful to compare the macroparticle size of AlTiN, AlTiN-Cu, AlTiN-Ni coatings, if the particle size is displayed in the figure.

Response: We added the MP (macroparticles) density is defined as fMPs = AMPs/Atotal 26, where AMPs and Atotal are the area of all the metallic MPs in the micrograph and the total area, respectively.

4) In Figure 4.

4-1. As my knowledge, the friction coefficient is deeply related to surface roughness. When Cu and Ni are doped, the surface roughness should be increased due to increase in macro particles. However, the friction coefficient is decreased after adding the Cu and Ni. I think that it is ironical results, so it is necessary to explain why the lubricant effect occurs.

Response:The Macroparticles of AlTiN coating consists of a Ti core- AlTiN shell. The macroparticles in AlTiN coating contribute to increased wear by collapse and release the abrasive fragments into the sliding contact. In corporation of Cu(Ni) into TiAlN coating, where Cu(Ni) forms as crystals rather than dissolves into TiAlN matrix due the Cu(Ni) is mutually immiscible with TiAlN and its nitride is thermally unstable. AlTiN-Cu(Ni) exhibits lower friction coefficient due the presence of Cu and Ni, Which have a lubricating effect during friction.

4-3. Authors mentioned that ‘The effective values are obtained by calculating the average values of the second parts’. If the second part is from 5 N to 18N, it is wrong. This part is still in error bar and they are not stable. Or if the second part is from 18N, authors need to show zoom-in image, because it is the proof of the author’s mention (line 137, The metallic~). And the difference between red and blue lines is not visible.

Response: We added images of a scratch on the surface.

5) In figure 8, Authors mentioned that ‘The good machining performance of AlTiN-Ni coating at high cutting speed is attributed to its relative low cutting force and high oxidation resistance’. However, in Figure 7. (a) and (b), AlTiN-Cu cutting force is lower than AlTiN-Ni and oxidation resistance of AlTiN is better than AlTiN-Ni. I think it is ironical results, so authors need explanation of relation between machining performance, cutting force and oxidation resistance.

 Response: The coating with adaptive lubricious usually have low friction coefficient and cutting force with elevating temperature, which can reduce adhesive wear and extend tool life. With the increase of cutting speed, more heat generated within the chip-tool contact zone cause a temperature increase on the cutting edge of the tools.Hence, the AlTiN-Ni coated tools behave higher tool lifetime than the AlTiN-Cu coated tool at high cutting speed, which is related to the balance of cutting force and oxidation resistance.

6) In figure 9 and 10, SEM images and EDS results of uncoated specimens and AlTiN coated specimens for investigating the wear behavior. However, the author’s paper is about AlTiN-Cu and AlTiN-Ni coatings, so I think that these results are irrelevant. And also if authors want to study about it, it is appropriate to add SEM images or EDS results of AlTiN-Cu and AlTiN-Ni coated specimens, as well.

 Response: We add the SEM images of AlTiN-Cu and AlTiN-Ni coatings.

7) In conclusion, Author mentioned the word ‘nanocomposite’. However, we cannot find any relevant data to explain the nano-scale. TEM images as my comment No.2 are still unclear for the word ‘nano’. And I don’t think that it is nanocomposite, because of XRD pattens and macroparticles in SEM images. If authors want to insist that it is nanocomposite, nano-scale analysis is obviously required, such as clear TEM images.

Response: We have deleted the word ‘nanocomposite’.

Round 2

Reviewer 2 Report

Minor

1) In figure 3, (a) and (c) may be replaced with (a) and (b).

2) In 149 sentence, Fig. 4. may be fixed as Fig. 5.

3) Author needs a full inspection of the figure once again.

Author Response

Dear Reviewer

        We sincerely thank the reviewer for your valuable feedbace that we have used to improve the quality of our manuscript. Those comments are helpful for us to revise and improve our paper.

  1  In figure 3, (a) and (c) may be replaced with (a) and (b)

Response: We agree that the figure 3, (a) and (c) should be replaced with (a) and (b). We fixed it

  2  In 149 sentence, Fig.4. may be fixed as Fig. 5.

Response: We agree that  the Fig.4 should be fixed as Fig. 5 in 149 sentence. We fixed it. 

  3  Author needs a full inspection of the figure once again

Response: We have a comprehesive inspection of the figure once again.